# Methionine Supplementation Affects Fecal Bacterial Community and Production Performance in Sika Deer (*Cervus nippon*)

**DOI:** 10.3390/ani13162606

**Published:** 2023-08-12

**Authors:** Yan Wu, Yongzhen Zhu, Xiaolan Guo, Xiaoxu Wang, Weitao Yuan, Cuiliu Ma, Xiaoli Chen, Chao Xu, Kaiying Wang

**Affiliations:** 1Innovation Center for Feeding and Utilization of Special Animals in Jilin Province, Institute of Special Animal and Plant Sciences, Chinese Academy of Agricultural Sciences, Changchun 130000, China; 2Research Center for Microbial Feed Engineering of Special Animals in Jilin Province, Institute of Special Animal and Plant Sciences, Chinese Academy of Agricultural Sciences, Changchun 130000, China

**Keywords:** apparent digestibility, methionine, fecal bacterial, short-chain fatty acids, sika deer

## Abstract

**Simple Summary:**

Sika deer is a rare livestock resource in China, and the main purpose of breeding is to obtain antlers that can be used as medicinal herbs. As an important limiting amino acid for ruminants, the dietary level of methionine greatly affects the utilization efficiency of protein in feed. In this study, we investigated the effects of different dietary methionine levels on antler-bearing sika deer by supplementing methionine. The results of the study showed that methionine supplementation improved the quality of Sika deer antlers and increased the efficiency of the animals utilizing acid detergent fiber and neutral detergent fiber in the feed. In addition, methionine supplementation altered the composition of free amino acids in the rumen fluid and the composition of fecal bacteria in sika deer.

**Abstract:**

Amino acid balance is central to improving the efficiency of feed protein utilization and for reducing environmental pollution caused by intensive farming. In previous studies, supplementation with limiting amino acids has been shown to be an effective means of improving animal nutrient utilization and performance. In this experiment, the effects of methionine on the apparent digestibility of nutrients, antler nutrient composition, rumen fluid amino acid composition, fecal volatile fatty acids and intestinal bacteria in antler-growing sika deer were investigated by randomly adding different levels of methionine to the diets of three groups of four deer at 0 g/day (CON), 4 g/day (LMet) and 6 g/day (HMet). Methionine supplementation significantly increased the apparent digestibility of organic matter, neutral detergent fiber (NDF) and acid detergent fiber (ADF) in the LMet group (*p* < 0.05). The crude protein and collagen protein of antlers were significantly higher in the LMet and HMet groups compared to the CON group and also significantly higher in the HMet group compared to the LMet group, while the calcium content of antlers was significantly lower in the HMet group (*p* < 0.05). Ruminal fluid free amino acid composition was altered in the three groups of sika deer, with significant changes in aspartic acid, citrulline, valine, cysteine, methionine, histidine and proline. At the phylum level, Firmicutes and Bacteroidetes were highest in the rectal microflora. Unidentified bacterial abundance was significantly decreased in the HMet group compared to the CON group. Based on the results of principal coordinate analysis (PCoA) and Adonis analysis, there was a significant difference in the composition of the intestinal flora between the CON and HMet groups (*p* < 0.05). At the genus level, compared with the CON group, the abundance of *Rikenellaceae_RC9_gut_group* and *Lachnospiraceae_UCG-010* in the LMet group increased significantly (*p* < 0.05), the abundance of *dgA-11_gut_group* in the HMet group decreased significantly (*p* < 0.05) and the abundance of *Lachnospiraceae_UCG-010*, *Saccharofermentans* and *Lachnospiraceae_NK3A20_group* increased significantly. Taken together, the results showed that methionine supplementation was beneficial in increasing the feed utilization efficiency and improving antler quality in sika deer, while affecting the composition of fecal bacteria.

## 1. Introduction

Intensive farming is a mode of feeding used in animal husbandry to maximize economic benefits, bringing high food and economic returns, but the associated wasting of resources and environmental pollution cannot be ignored [1,2]. Animal nutritionists hope to reduce protein feed waste and reduce nitrogen excretion by specifying optimal amino acid levels in feed [3]. Methionine is the first limiting amino acid in ruminants and affects the efficiency of feed utilization. Its supplementation can significantly improve the absorption of other amino acids in the intestine because the ratio of amino acids becomes more balanced and the antagonism between various amino acids is reduced [4]. This improves the oxidative damage of intestinal epithelial cells and enhances the immune barrier function of the intestine, maintaining intestinal health and nutrient absorption capacity [5].

Deer antler has been an important Chinese herbal medicine since ancient times. Many studies have shown that antlers are rich in active ingredients, which can replenish blood, invigorate energy and improve sexual function [6,7]. They are very rich in peptides and proteins which are considered to be important components for promoting tissue repair, antioxidant production and neuroprotection [8]. Antler polysaccharide, one of the important components of antler, is believed to prevent osteoporosis and enhance immunity [9]. However, the long breeding cycle, high feed cost and the ability to harvest antlers only once or twice a year make antlers expensive. Furthermore, the specific mechanism of the active ingredients of antlers in treating diseases is not clearly understood, which limits the use of antlers and the breeding of sika deer.

Gut microbes have been shown to influence host nutrient utilization, immune function regulation, tissue and organ inflammation, disease development and treatment through their metabolites and derivatives [10,11]. The composition and function of the intestinal microbiota are extremely complex and the differences in their composition depend mainly on environmental changes and the role of host gene expression. During long-term co-evolution with the host, the colonization of the microbiota has stabilized, and its function has improved [12,13]. Microbial–host interactions can provide a suitable living environment for each other, and the host can provide a more stable microenvironment for microorganisms to inhabit and utilize nutrients for their own use and the metabolites produced by microorganisms through fermentation, such as short-chain fatty acids (SCFA) can be efficiently used by the host, while a stable microbial composition can also effectively resist the invasion of pathogenic microorganisms [14].

In our previous study, it was shown that methionine supplementation significantly improved the antioxidant and immune levels of sika deer, which was an effective measure to reduce environmental and physiological stresses in these timid and sensitive deer. In the current study, we further evaluate the effects of methionine on production and on rectal bacteria and bacterial fermentation to refine the assessment of the appropriate level of methionine supplementation.

## 2. Materials and Methods

### 2.1. Experimental Design and Diet

The animals selected for this experiment were 12 four-year-old healthy male sika deer, with a mean weight of 103.2 ± 16.7 kg, with no significant differences between groups. All animal procedures were approved and authorized by the Animal Ethics Committee and Animal Protection and Utilization Committee of the Chinese Academy of Agricultural Sciences.

The 12 animals were randomly assigned to three treatment groups and each deer was fed in a separate enclosure. The experiment was preceded by a one-week pre-experiment and consisted of a four-week dietary treatment in June 2021. The deer in each group were fed a corn silage and concentrate based diet with a dry matter content of 35% to 65% [15]. Methionine was randomly added to the basal diet at 0 g/day (CON), 4 g/day (LMet) or 6 g/day (HMet). Methionine is crystalline *DL*-Met with 99% purity (Degussa, Frankfurt, Germany). All animals were fed a 2.8 kg diet daily on a dry matter basis, split equally between a 4 a.m. and 5 p.m. feeding, with free access to drinking water. Every daily feed was completely consumed.

### 2.2. Sample Collection

The fecal samples were collected by a partial collection method every day before morning feeding for three consecutive days before the end of the four-week experiment. Four fresh fecal samples were collected from each pen and each sample was about 100 g. After removing the hair and gravel, the collected fecal samples were divided into two parts, with one part sprayed with 10% dilute sulfuric acid to fix nitrogen, dried to constant weight at 65 °C, crushed and passed through a 40-mesh sieve and air-dried to determine the crude protein level [16]. The other part was sterilized in the oven at 85 °C for two hours, dried at 65 °C to constant weight and then crushed through an 18-mesh sieve to determine the fiber content [17].

At the end of the experiment, the animals were anesthetized with chlorpromazine hydrochloride injection applied to the gluteal muscles, at 2.0 mL per 100 kg body weight before the morning feed. The antlers were cut with a sterilized saw, weighed and dried at 65 °C to constant weight, and the whole intact antlers were crushed and prepared for use. At the same time, 200 mL of ruminal fluid was obtained from the rumen by nasogastric tube sampler, with the first 100 mL of the rumen fluid discarded to avoid saliva contamination and a 20 g fecal sample obtained per rectum for microbiological sequencing [18]. These samples were rapidly transferred to liquid nitrogen and stored at −80 °C pending further analysis.

### 2.3. Determination of Apparent Nutrient Digestibility and Nutrient Content

Dry matter (DM) content was determined with reference to national recommended standards GB/T 6435-2014 and crude protein (CP) content determined with reference to GB/T 6432-2018 using Kjeldahl’s method. Ether extract (EE) content was determined using the Soxhlet fat extraction method with reference to GB/T 6433-2006. The neutral detergent fiber (NDF) and acid detergent fiber (ADF) content were determined using the Van Soest method with reference to GB/T 20806-2006. The polysaccharides and collagen were determined using Gong’s method [19,20]. Calcium content was determined according to GB/T 13885-2003 and the phosphorus content was determined according to GB/T 6437-2018.

The apparent digestibility of amino acids (%) was determined as 100 − (100 × *A*/*A*_1_ × *B*_1_/*B*), where *A* was the percentage of 2 mol/L hydrochloric acid insoluble ash in the diet, *A*_1_ was the percentage of 2 mol/L hydrochloric acid insoluble ash in the feces, *B*_1_ was the percentage of amino acid in the feces and *B* was the percentage of amino acids in the diet. All national standard test methods can be obtained from the website http://www.moa.gov.cn/ (accessed on 10 September 2021).

### 2.4. Determination of Free Amino Acids in Rumen Fluid

The rumen fluid was centrifuged at 16,000× *g* for 15 min at 4 °C, then the supernatant was mixed with trichloroacetic acid (10%) at 1:1 and recentrifuged at 16,000× *g* for 15 min at 4 °C to precipitate the rumen fluid protein, then filtered into a 2 mL injection bottle. The free amino acid content of rumen fluid was determined by ion exchange chromatography using an L-8900 amino acid auto analyzer [21] (Hitachi Technology, Tokyo, Japan).

### 2.5. Determination of Fecal Short-Chain Fatty Acids

For GC–MS detection, 600 μL of the standard was added to 25 μL of 4-methylvaleric acid at a final concentration of 500 μM as the internal standard, mixed well and added into the injection vial with a sample volume of 1 μL and a splitting ratio of 10:1. The samples were thawed on ice and 30 mg of the samples were taken in 2 mL glass centrifuge tubes and mixed for 2 min by shaking with 900 μL of 0.5% phosphoric acid to resuspend. The sample was then centrifuged at 22,000× *g* for 10 min and 800 μL of supernatant had an equal amount of ethyl acetate added for extraction. It was shaken and mixed for 2 min, centrifuged at 22,000× *g* for 10 min and 600 μL of the upper organic phase added with 25 μL of 4-methylvaleric acid at a final concentration of 500 μM as the internal standard, mixed well and added into the injection vial for GC–MS detection with a sample volume of 1 μL and a splitting ratio of 10:1. The samples were separated on a 30 m × 0.25 mm ID × 0.25 μm DB-WAX capillary column gas chromatography system (Agilent Technologies Inc., Santa Clara, CA, USA). A quality control (QC) sample was set up for each experimental sample in the sample cohort to test and evaluate the stability and reproducibility of the system. Mass spectrometry was carried out using a 7890A/5975C gas–mass spectrometer (Agilent). The peak areas and retention times of the chromatograms were extracted using MSD ChemStation software Version B.08.00 (Agilent). The calibration curves were plotted and the contents of short-chain fatty acids in the samples were calculated.

### 2.6. Microbial Sequencing and Analysis

The DNA was extracted using the cetyltrimethylammonium bromide method [22]. After testing the purity and concentration, it was diluted to 1 ng/uL using sterile water and used as a template for polymerase chain reaction (PCR) amplification. The PCR amplification was performed using specific primers with Barcode, Phusion^®^ High-Fidelity PCR Master Mix with GC Buffer (New England Biolabs, Ipswich, MA, USA) and high-performance, high-fidelity enzymes (New England Biolabs). After detection by agarose gel electrophoresis at 2% concentration, the qualified PCR products were purified with magnetic beads and quantified by enzyme labeling, using recovery kits (Qiagen, Redwood City, CA, USA). Libraries were constructed using a TruSeq DNA PCR-Free Sample Preparation Kit (Illumina, San Diego, CA, USA). The libraries were quantified by quantum bit (Qubit) and real-time quantitative polymerase chain reaction (Q-PCR) and then sequenced using NovaSeq6000 (Illumina, San Diego, CA, USA) [23].

The reads of each sample were spliced using FLASH version 1.2.7 (http//ccb.jhu.edu/software/FLASH/ accessed on 15 September 2021) after truncating the barcode and primer sequences and the spliced sequences were the original Tags data (raw tags). The raw tags obtained by splicing went through a strict filtering process to get high quality tags data (clean tags) and after the chimeric sequences were removed, the final valid data were obtained. The Uparse algorithm version 7.0.1001 (http://www.drive5.com/uparse/ accessed on 15 September 2021) was used to cluster the effective tags of all samples and the sequences were clustered by default parameters with 97% consistency into operational taxonomic units (OTUs). The Chao1, Shannon, Simpson, and ACE indices and Unifrac distances were calculated using Qiime software version 1.9.1 and an alpha diversity index intergroup variance analysis and PCoA plotting were performed using R software version 2.15.3. The PCoA analysis was performed using weighted correlation network analysis (WGCNA) and the “stats” and “ggplot2” R packages. Adonis analysis was performed using the Adonis function of the “vegan” R package.

### 2.7. Data Analysis

The analysis process used the analysis software SPSS Statistics 26 (IBM-SPSS Inc., Chicago, IL, USA). The Shapiro–Wilk test was used to determine whether the data conformed to a normal distribution and the parametric test was used to confirm that the data conformed to a normal distribution. Homogeneity of variance was determined using the F-test and one-way analysis of variance (ANOVA) with post-hoc multiple comparisons using the Tukey test was used when homogeneity of variance was met and the *t*-test was used when not met, with a significant value of *p* < 0.05. Data are shown as the mean ± standard error.

## 3. Results

### 3.1. Methionine Supplementation Improves Apparent Digestibility of Nutrients

The experimental results in Table 1 show that methionine supplementation significantly improved the apparent digestibility of organic matter, NDF and ADF in the LMet group compared to the other two groups (*p* < 0.05), while the digestibility of the HMet group was improved but not significantly. The addition of methionine tended to increase the apparent digestibility of crude protein compared to the CON group (*p* < 0.1). There was no significant difference in the apparent digestibility of the ether extract and dry matter among the three groups.

### 3.2. Quality of Sika Deer Antlers Is Enhanced by Methionine Supplementation

The results in Table 2 show that the crude protein and collagen contents of the antlers in the three groups were significantly different (*p* < 0.05), with the highest in the HMet group and the lowest in the CON group. The calcium content of the three groups gradually decreased with the addition of methionine, with the CON group being significantly higher than the HMet group (*p* < 0.05). However, there were no significant differences in the ether extract, polysaccharide and phosphorus levels among the three antler groups.

### 3.3. Alteration of Free Amino Acids in the Rumen Fluid of Sika Deer

Methionine supplementation significantly altered the free amino acid composition of the rumen fluid of the antler-bearing sika deer, as seen in Table 3. In the essential amino acid fraction, the methionine content of the LMet and HMet groups increased significantly and varied linearly with dietary methionine levels, along with a significant increase in valine content and a decrease in histidine content in the LMet group (*p* < 0.05). In the non-essential amino acid fraction, proline and cysteine were significantly higher and citrulline was significantly lower in the LMet and HMet groups compared to the CON group, as well as a significant increase in aspartic acid in the LMet group (*p* < 0.05). There was no significant difference among the three groups of amino acid derivatives.

### 3.4. Effect of Methionine Supplementation on Short-Chain Fatty Acids in the Feces of Sika Deer

The results in Table 4 show that the effect of methionine supplementation on fecal volatile fatty acids was not significant. The HMet group had the highest levels of SCFA, while the CON group had the lowest, but the differences in total volatile fatty acids, as well as levels of all types of short-chain fats among the groups, were not significant (*p* > 0.05).

### 3.5. Bacterial Composition of the Feces of Sika Deer and the Characteristic Bacteria of Each Group

The PCR-free library construction was based on the Illumina Nova sequencing platform, followed by paired-end sequencing. By splicing reads, the amount of valid data for quality control reached 55,270. Sequences were clustered into operational taxonomic units (OTUs) with 97% identity and a total of 4553 OTUs were obtained after removing the data results of archaea, unknowns and no blast hits. The OTU sequences were then species annotated with the Silva138 database.

The analysis showed that at the gate level, Firmicutes in CON at 51.5 ± 2.3%, LMet at 53.0 ± 2.0% and HMet at 57.1 ± 2.2% and Bacteroidetes in CON at 30.1 ± 1.1%, LMet at 31.3 ± 2.1% and HMet at 33.1 ± 5.6% had the highest abundance, accounting for more than 80% of the bacterial abundance in feces. At the genus level, *Oscillospiraceae_UCG-005* had the highest abundance in CON at 15.5 ± 2.6%, LMet at 14.7 ± 1.6% and HMet at 18.7 ± 3.1%, accounting for more than 15% of the total. The *Rikenellaceae_RC9_gut_group* in CON at 5.3 ± 0.2%, LMet at 6.2 ± 0.2% and HMet at 7.6 ± 1.2%; *Treponema* in CON at 4.0 ± 1.3%, LMet at 4.0 ± 0.8% and HMet at 2.4 ± 0.5%; *Bacteroides* in CON at 4.7 ± 0.1%, LMet at 4.2 ± 0.5% and HMet at 5.4 ± 0.9%; *Alistipes* in CON at 3.0 ± 0.2%, LMet at 3.4 ± 0.2% and HMet at 3.9 ± 0.5% and *Christensenellaceae_R-7_group* in CON at 2.6 ± 0.4%, LMet at 3.2 ± 0.1% and HMet = 2.4 ± 0.2%, were other microorganisms widely present in the feces of sika deer, as shown in Figure 1.

By comparing the α diversity of different groups, the results in Figure 2, show that the Shannon, Chao1 and ACE indices were significantly lower in the HMet group than in the CON and LMet groups (*p* < 0.05), but there was no significant difference in the Simpson index among the three groups. The PCoA results in Figure 3 and Adonis analysis results in Table 5 showed that based on the Bray–Curtis distance, binary Jaccard distances, weighted uniFrac distance and unweighted uniFrac distances matrix results, the composition of the flora of the HMet and CON groups were significantly separated (Adonis: *p* < 0.05), while there was no significant difference between the LMet group and the other two groups (*p* > 0.05).

By further comparison, other significantly different groups of bacteria among groups at the phylum and genus level were found in Figure 4. At the phylum level, the abundance of unidentified_bacteria were significantly decreased in the HMet group compared to the CON group. At the genus level, the abundance of *Rikenellaceae_RC9_gut_group* and *Lachnospiraceae_UCG-010* was significantly upregulated in the LMet group compared to the CON group, *dgA-11_gut_group* was significantly downregulated, *Lachnospiraceae_UCG-010*, *Saccharofermentans* and the *Lachnospiraceae_NK3A20_group* were significantly upregulated in the LMet group compared to the CON group (*p* < 0.05).

## 4. Discussion

The apparent digestibility of organic matter, NDF and ADF increased significantly in the LMet group under the experimental conditions, which was consistent with previous predictions for changes in nutrient utilization. A previous study found changes in the abundance of a variety of rumen bacteria associated with cellulose degradation, especially in the LMet group, which showed a significant increase, improving the animal’s ability to utilize crude fiber components in the feed [15]. It has also been shown that methionine supplementation had a positive effect on the fermentation rate of NDF in the rumen of heifers, which facilitated the utilization of cellulose by the animals [24]. Protein and peptide nutrients in feed are not only the main way for animals to take in amino acids, but also the only way for rumen microorganisms to obtain amino acids and different animals have different levels of need for different amino acids [25,26], while microorganisms also have certain preferences for amino acid utilization [27,28]. We believe that methionine supplementation promoted the reproduction and function of some cellulose-degrading bacteria, so the role of methionine on rumen microorganisms and the mechanism of action will be further explored in subsequent studies to clarify the interaction between methionine, microbial metabolism and host development. In addition, a trend in the apparent digestibility of crude protein (*p* < 0.1) was observed with both the LMet and HMet groups outperforming the CON group, which was attributed to a more balanced amino acid ratio in the feed. Methionine as a limiting amino acid significantly affects the absorption and utilization of amino acids in ruminants and supplementation with methionine can effectively reduce the degree of amino acid imbalance in corn–soybean meal-based diets and promote protein utilization [29].

During the antler growth phase, sika deer need more energy intake to ensure cell proliferation and differentiation, and protein and peptides, as the main active ingredients for the medicinal effects of antlers, need sufficient protein intake to ensure medicinal value [30,31]. As such, the efficient utilization of protein and energy materials may mean that sika deer will show better productive performance. By further analysis of the nutritional composition of the antlers, we found that supplementation with methionine did significantly increase the crude protein and collagen content of plum deer antlers, while significantly decreasing the calcium content of the antlers. As the most abundant active ingredients in deer antler, antler protein and antler polypeptide have various medical effects in clinical practice, including improving immune function and antioxidant levels in animals [32,33], resisting acetaminophen-induced nephrotoxicity [34], improving neurological damage and cognitive dysfunction [35], repairing vascular and epithelial tissue damage and accelerating wound healing [36]. Calcium deposition as a landmark process of antler aging and the calcium and collagen content of antler, showed a significant linear relationship with the freshness of the antler, and the higher the degree of ossification the corresponding decrease in medicinal value [37,38]. The effect of methionine supplementation on antler composition was due to the improved nutrient utilization and the ability of sika deer to obtain more energy and amino acids for antler growth and protein deposition. We believe that methionine levels affect the expression levels of hormones associated with antler growth, based on changes in alkaline phosphatase associated with the ossification process detected in a previous study, but this is not direct evidence and further investigation is needed.

Proteins, peptides, amino acids and other nutrients consumed by ruminants are often altered in their amino acid composition and ratio before entering the intestine to be absorbed and used by the organism. This is because bacteria, fungi, protozoa and other microorganisms living in the rumen first carry out vital activities for these nitrogenous substances as nitrogen sources for the synthesis of microbial proteins, producing various amino acid metabolites through transamination and deamination [39].

Previous studies have shown that supplementation with exogenous amino acids or diets with different amino acid compositions can alter the composition characteristics and fermentation properties of rumen microorganisms, affecting the amino acid composition that actually enters the intestine of ruminants [40]. Our study aimed to investigate the differences in amino acid composition entering the intestinal tract of sika deer at different methionine levels in feed by assessing the free amino acid composition of rumen fluid. Under the conditions of this experiment, the content of methionine, valine, cysteine, and proline in the rumen fluid of sika deer increased significantly after supplementation with methionine, while the content of histidine and citrulline decreased significantly. This result indicated that the rumen microorganisms had undergone amino acid utilization and metabolism, which resulted in a change in the composition of the amino acids entering the intestine for each group. It has also been shown that the free amino acid composition of rumen fluid in buffaloes changed significantly when methionine supplements were used [41]. We speculate that differences in the composition of amino acid nutrients in the enteric chyme will also affect the composition of the intestinal bacteria and the fermentation process. To test this conjecture, the bacterial composition of the rectum and the results of bacterial fermentation were analyzed by 16s rRNA high-throughput sequencing and chromatograph analysis.

Numerous studies have demonstrated the important role of the gut microbiome in animal health and disease and the microbiota is highly sensitive to environmental factors, with changes in the cecum and fecal microbiota being significantly associated with changes in diet structure [42]. The interaction between the gut microbiota and the host is reflected not only in the influence of the host’s diet and other lifestyle habits on the microbes, but also in the metabolites produced by the microbes during the fermentation of food and their own metabolism. This in turn can affect the host’s own immune function and inflammatory response and alter the host’s neurocognitive function, cellular aging and apoptosis, as well as other processes [43,44]. Microorganisms living in the large intestine of ruminants are able to improve the assistance of cellulose degradation. Short-chain fatty acids (SCFA) are important fermentation products which provide the animal with a source of energy and act on the cells of the intestinal wall or enter the body circulation via the capillaries [45,46]. Studies have shown that SCFA can help protect against pathogenic microorganisms by improving intestinal damage and maintaining intestinal barrier function [47]. In our study, we observed some variation in the content of SCFA in each group, with the highest total SCFA content in the HMet group and the lowest in the CON group, consistent with the fact that the level of acetic acid also showed a trend of positive correlation with the level of feed methionine. By performing alpha diversity analysis of the intestinal flora, we found that the Shannon, Chao1 and ACE indices of the HMet group showed significant decreases relative to the CON and LMet groups. Alpha diversity reflects the richness and diversity of the bacterial flora, and higher richness and diversity of the flora tend to indicate a more robust intestinal microbial composition and higher resistance to pathogenic microbial attack [48]. This seems to imply that high levels of methionine supplementation are detrimental to the stability of the intestinal flora and the health of the host, but this finding is contrary to previous results of improved immune and antioxidant function in the HMet group. The differences in the composition of the intestinal microorganisms of the three groups were further analyzed and the results of PCoA and Adonis analyses showed a significant separation of the flora between the HMet and CON groups (*p* < 0.05). By analyzing the differences in bacterial changes at the phylum level and genus level, the results showed that the HMet group had many unidentified bacteria with significantly decreased abundance. We hypothesize that these unidentified bacteria might be pathogenic bacteria in the host and that therefore the HMet group still showed the best production performance and better nutrient utilization efficiency despite the decreased diversity of the flora. In support of this hypothesis, it has been shown that the unidentified bacteria limit the production of SCFA in the rumen [49].

The abundance of several bacteria were altered at the genus level. *Lachnospiraceae_UCG-010* was considered in previous studies as a beneficial bacterium in the large intestine that can provide positive effects on intestinal development and health by degrading plant fiber to produce SCFA [50] and *Saccharofermentans* is a typical acetic acid-producing bacterium that plays a potential role for animals in nutrient utilization [51]. *Lachnospiraceae_NK3A20_group* is a butyrate-producing bacterium that is beneficial to intestinal health and has also been found to be significantly negatively correlated with residual feed intake in animals, closely related to linoleic acid metabolism and lysine degradation in ruminants and able to improve metabolic levels in animals [52]. *Rikenellaceae_RC9_gut_group* and *Lachnospiraceae_UCG-010*, which appear to be upregulated in the HMet group, are also considered to facilitate the utilization of crude fiber feed and the production of SCFA in animals [53,54]. These results suggested that changes in fecal microbial response to methionine supplementation were positive.

## 5. Conclusions

Antler quality is an important indicator of the benefits of deer breeding, and our research proves that supplementation with appropriate methionine can effectively improve the production performance of sika deer and provide more economic benefits for breeding. Methionine supplementation improved the efficiency of feed utilization by the animals, owing to a more balanced ratio of amino acids in the diet, which reduced the waste of protein resources, and because of the increased abundance of beneficial bacteria in the large intestine, which promoted the degradation of cellulose. This provided the necessary energy and protein supply for the rapid growth of antlers in antler-growing sika deer. We also found that differences in the amino acid composition of the diets significantly affected the utilization of protein by rumen microorganisms, resulting in a change in the actual amino acid composition that enters the intestine of the sika deer. Whether this change has a positive or negative effect on the animal, however, cannot be determined at this time. Overall, dietary methionine supplementation improved the nutrient composition of antlers while increasing the feed utilization efficiency and positively influencing the large intestine bacterial composition of sika deer. Considering the product benefits and costs, we recommend 4 g/day as a supplemental dose of methionine for antler-bearing sika deer.

## Figures and Tables

**Figure 1 animals-13-02606-f001:**
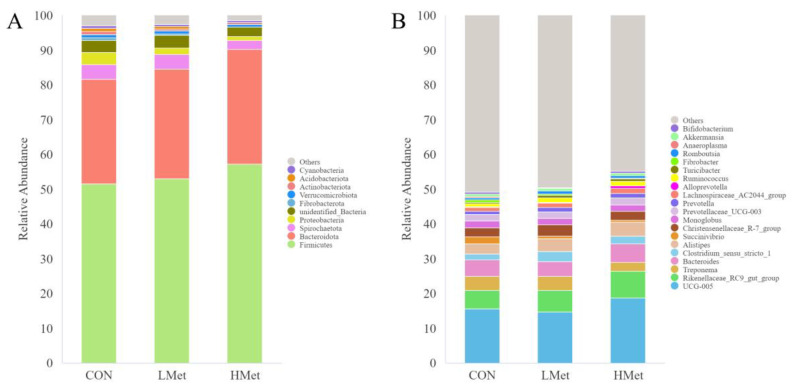
Bacterial composition in the feces of sika deer at the phylum (**A**) and genus (**B**) levels.

**Figure 2 animals-13-02606-f002:**
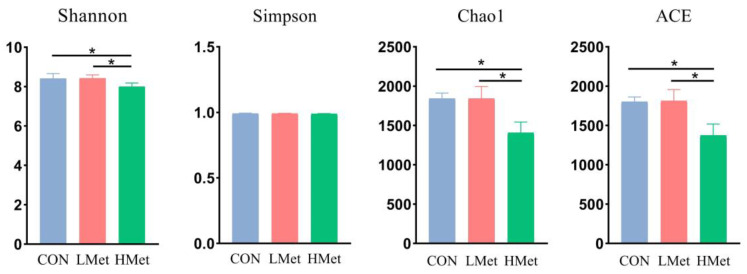
Comparisons of the alpha diversity of the bacteria in the feces of sika deer. * indicates *p* < 0.05.

**Figure 3 animals-13-02606-f003:**
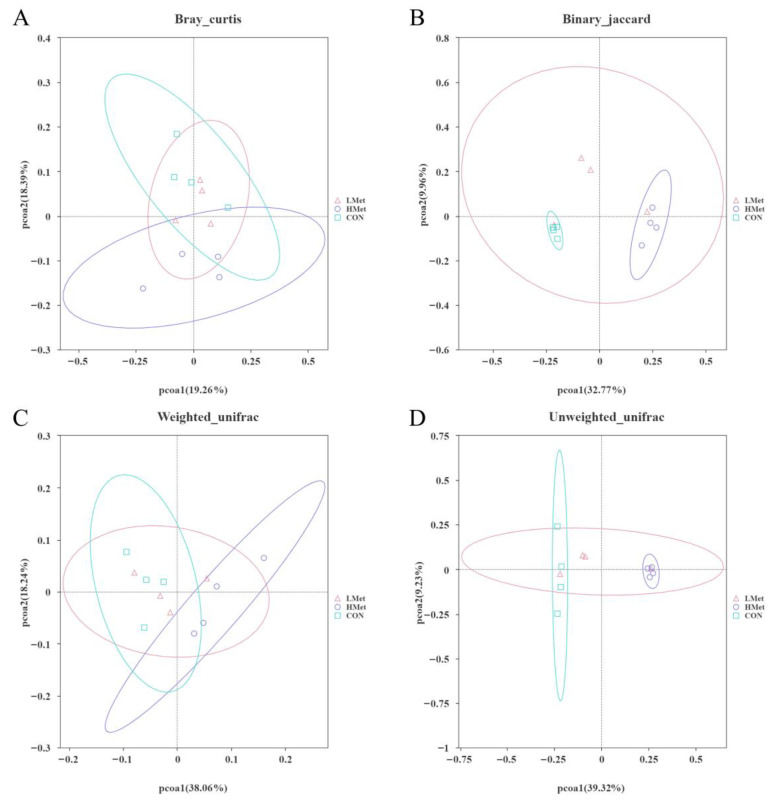
Comparisons of the bacterial communities in the gut of sika deer. Principal coordinate analyses based on the Bray Curtis distance (**A**), binary Jaccard distances (**B**) weighted UniFrac distance (**C**) and unweighted UniFrac distance (**D**).

**Figure 4 animals-13-02606-f004:**
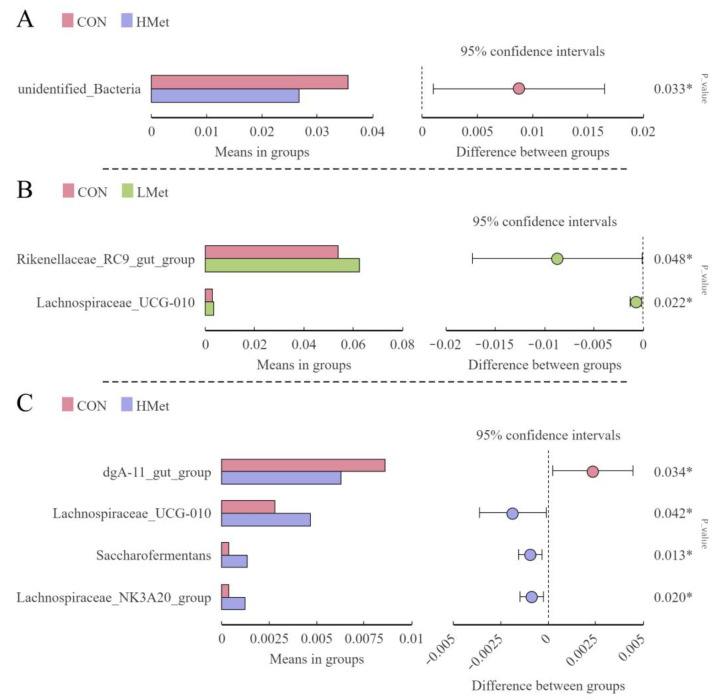
T-test bar plots show significant differences in the relative abundance of bacteria at the phylum (**A**) and genus (**B**,**C**) levels. Comparisons without significant differences are not shown. * indicates *p* < 0.05.

**Table 1 animals-13-02606-t001:** Apparent digestibility of nutrients in three groups.

Item (%)	CON	LMet	HMet	*p*-Value
Crude protein	80.23 ± 1.41	84.54 ± 1.04	82.01 ± 1.24	0.09
Ether extract	88.79 ± 1.33	90.80 ± 0.60	89.33 ± 1.72	0.55
Dry matter	77.91 ± 1.03	80.68 ± 1.49	76.00 ± 1.52	0.10
Organic matter	73.79 ^b^ ± 1.55	82.55 ^a^ ± 1.38	76.23 ^b^ ± 1.25	<0.01
NDF	79.02 ^b^ ± 0.58	82.96 ^a^ ± 0.49	79.42 ^b^ ± 0.94	<0.01
ADF	55.76 ^b^ ± 1.89	65.47 ^a^ ± 1.42	58.44 ^b^ ± 1.44	<0.01

ADF, acid detergent fiber; NDF, neutral detergent fiber. Note: values with different letter superscripts represent a significant difference (*p* < 0.05), while those with the same letter or no letter superscripts represent no significant difference.

**Table 2 animals-13-02606-t002:** Effect of methionine supplementation on antler quality (air-drying foundation).

Item (%)	CON	LMet	HMet	*p*-Value
Crude protein	62.01 ^c^ ± 0.55	65.05 ^b^ ± 0.66	69.83 ^a^ ± 0.31	<0.01
Ether extract	2.85 ± 0.09	3.10 ± 0.04	2.96 ± 0.05	0.15
Collagen protein	21.45 ^b^ ± 0.49	24.89 ^a^ ± 0.61	25.93 ^a^ ± 0.23	<0.01
Polysaccharide	0.84 ± 0.02	0.85 ± 0.04	0.89 ± 0.03	0.57
Ca	7.67 ^a^ ± 0.28	6.81 ^ab^ ± 0.30	6.53 ^b^ ± 0.20	0.03
P	5.97 ± 0.06	6.03 ± 0.08	5.77 ± 0.14	0.21

Note: values with different letter superscripts represent a significant difference (*p* < 0.05), while those with the same letter or no letter superscripts represent no significant difference.

**Table 3 animals-13-02606-t003:** Free amino acid composition of the rumen fluid among the three groups.

Item (nmol/mL)	CON	LMet	HMet	*p*-Value
Phosphoserine	1.09 ± 0.10	0.83 ± 0.04	0.84 ± 0.05	0.05
Aspartic acid	3.22 ^b^ ± 0.15	5.10 ^a^ ± 0.59	4.38 ^ab^ ± 0.43	0.03
Threonine	2.35 ± 0.27	2.15 ± 0.31	2.43 ± 0.26	0.77
Serine	2.42 ± 0.11	2.80 ± 0.11	2.60 ± 0.06	0.06
Glutamate	10.06 ± 0.51	10.27 ± 0.74	10.52 ± 0.89	0.85
α-Aminoadipate	0.10 ± 0.005	0.10 ± 0.007	0.12 ± 0.008	0.32
Glycine	3.12 ± 0.14	3.64 ± 0.73	3.17 ± 0.46	0.35
Alanine	4.92 ± 0.03	5.38 ± 0.19	5.03 ± 0.18	0.15
Citrulline	0.69 ^a^ ± 0.06	0.37 ^b^ ± 0.03	0.46 ^b^ ± 0.04	<0.01
α-Aminobutyric acid	0.18 ± 0.01	0.19 ± 0.01	0.20 ± 0.01	0.59
Valine	2.27 ^b^ ± 0.13	3.00 ^a^ ± 0.22	2.85 ^ab^ ± 0.14	0.03
Cysteine	0.24 ^b^ ± 0.01	0.37 ^a^ ± 0.02	0.42 ^a^ ± 0.01	<0.01
Methionine	1.05 ^b^ ± 0.06	1.40 ^a^ ± 0.05	1.52 ^a^ ± 0.04	<0.01
Cystathionine	0.24 ± 0.01	0.28 ± 0.03	0.25 ± 0.02	0.52
Isoleucine	2.06 ± 0.10	2.10 ± 0.13	2.11 ± 0.06	0.95
Leucine	2.32 ± 0.07	2.36 ± 0.09	2.42 ± 0.07	0.72
Tyrosine	1.39 ± 0.10	1.29 ± 0.13	1.30 ± 0.02	0.75
Phenylalanine	1.17 ± 0.10	1.24 ± 0.13	1.12 ± 0.05	0.70
Beta alanine	0.94 ± 0.03	1.02 ± 0.05	0.96 ± 0.03	0.43
γ-Aminobutyric acid	0.18 ± 0.01	0.16 ± 0.02	0.18 ± 0.01	0.82
Ethanolamine	0.63 ± 0.03	0.57 ± 0.08	0.56 ± 0.05	0.71
NH_3_	100.66 ± 6.84	97.94 ± 4.21	91.57 ± 3.28	0.45
Hydroxylysine	0.46 ± 0.01	0.45 ± 0.04	0.46 ± 0.02	0.87
Ornithine	0.55 ± 0.02	0.54 ± 0.03	0.51 ± 0.01	0.61
Lysine	5.38 ± 0.50	6.33 ± 0.38	5.99 ± 0.82	0.34
Histidine	0.28 ^a^ ± 0.02	0.19 ^b^ ± 0.01	0.23 ^ab^ ± 0.01	<0.01
Proline	2.83 ^b^ ± 0.10	4.20 ^a^ ± 0.21	4.08 ^a^ ± 0.18	<0.01

Note: values with different letter superscripts represent a significant difference (*p* < 0.05), while those with the same letter or no letter superscripts represent no significant difference.

**Table 4 animals-13-02606-t004:** Composition of short-chain fatty acids in the feces.

Item (μg/g)	CON	LMet	HMet	*p*-Value
Acetic acid	1125.12 ± 60.53	1194.64 ± 60.45	1223.39 ± 78.63	0.58
Propionic acid	370.50 ± 45.32	362.41 ± 19.77	396.28 ± 38.02	0.79
Isobutyric acid	26.50 ± 2.80	27.45 ± 3.95	26.91 ± 3.10	0.98
Butyric acid	348.83 ± 21.62	327.24 ± 19.32	325.04 ± 22.70	0.69
Isovaleric acid	18.10 ± 3.36	19.12 ± 2.42	19.27 ± 3.70	0.96
Valeric acid	48.42 ± 2.88	44.39 ± 2.52	48.28 ± 1.58	0.43
Hexanoic acid	3.50 ± 0.45	3.22 ± 0.52	3.33 ± 0.41	0.91
Total SCFA	1941.00 ± 87.44	1978.48 ± 79.41	2042.51 ± 82.78	0.69

**Table 5 animals-13-02606-t005:** Adonis analysis of the bacterial communities in the feces of sika deer.

Group	Bray Curtis		Weighted UniFrac		Binary Jaccard		Unweighted UniFrac	
	R^2^	*p*	R^2^	*p*	R^2^	*p*	R^2^	*p*
CON vs. LMet	0.145	0.446	0.130	0.625	0.192	0.124	0.192	0.079
CON vs. HMet	0.230	0.056	0.359	0.033	0.417	0.030	0.486	0.024
LMet vs. HMet	0.175	0.119	0.240	0.072	0.241	0.061	0.293	0.066

## Data Availability

Relevant data for this article can be obtained by contacting the author.

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
