# Peer review of "Methionine Supplementation Affects Fecal Bacterial Community and Production Performance in Sika Deer (Cervus nippon)"

_animals, 2023, doi:10.3390/ani13162606_

Round 1
Reviewer 1 Report
The work rigorously describes how Methionine supplementation affects the composition and production of bacteria at the fecal level as well as improving the digestive capacity of Sika deer and improving the quality of antler, in terms of collagen and protein content. Thanks to the writing of the work I make only a few simple comments
Line 24-25 should be improved the sentence since the part describing Firmicutes and Bacteroidetes seems incomplete
Row 53 What do you mean by "improve osteoporosis"?
In the part of experimental design and diet, I suggest to describe the period of the year in which the test was made and improve the description of the diet provided also in terms of composition as well as the observed consumption
Table 1,2 3 , even if described in the text I suggest to specify mean+ standard error in the description of the legend, as well as I suggest to explain the meaning of the different letters on the mean values when statistically different
Table 2, what do you mean by item (%) ? is the % on dry matter ? specify better
Figure 2 and 4 can you explain in the caption, the meaning of the asterisks
Line 372 can you explain and support with citation to the judgment on the pathogenic of unidentified bacteria ?
Author Response
Dear professor
Thank you very much for your patient and careful review of my paper and your pertinent suggestions, which have helped me a lot in optimizing my paper.
In response to your comments, the response is as follows
- Thank you very much for your suggestions, which I have added.
- The reference to the medical effects of antler polysaccharides, which have a therapeutic effect on osteoporosis, may not have been worded strictly and has been modified.
- The experimental period has been added, but the diet description has not been added, because of concerns about too many charts and graphs in the article, and because our previous article has been cited so that the composition of the diet can be clearly seen. In addition, the amount of food taken has been described and the food was completely consumed by the animals.
- Changes have been made to identify under each table
- The percentage sign indicates the proportion, which has been indicated as air-dried basis.
- It has been annotated to say what the asterisk means.
- It is not possible to determine the pathogenicity of unclassified bacteria, but previous studies have reported negative effects on the production of short-chain fatty acids, and these bacteria are negatively correlated with some SFAs, which seems to be detrimental to the production and health of animals. The fact that the animals were not in poor health and that HMet's immune and antioxidant levels were elevated is of course not direct evidence of their pathogenicity, so we are only speculating and do not express a definitive statement in the paper. Unfortunately, the sika deer are very valuable and it is difficult to obtain direct evidence because of the difficulties of slaughtering and obtaining intestinal samples; we may follow up with metabolomic analysis of blood and fecal samples, and hopefully we will find something.
In addition to your suggestions, other reviewers' suggestions have also been revised in the paper.
Finally, thanks again for your help!
Best wishes.

Reviewer 2 Report
Good afternoon!
Thank you for sending the manuscript to Animals magazine. In general, evaluating the work positively, it is necessary to note some inaccuracies that need to be eliminated:
1. What is the reason for the different temperature of fecal drying for the determination of protein and fiber?
2. I think it's better to start chapters with a sentence, not the names of tables.
3. Specify were methionine supplements in protected form or native ?
4. The conclusions do not contain specifics. It is necessary to rewrite briefly and concisely which option the authors recommend.
5. The list of references should be brought in line with the requirements of the journal.
6. Other comments on the text.

Author Response
Dear professor
Thank you very much for your patient and careful review of my paper and your pertinent suggestions, which have helped me a lot in optimizing my paper.
In response to your comments, the response is as follows
- Usually air-dried samples are prepared at 65°C. However, since some microorganisms capable of degrading fiber are present in faces and some fermented feeds, it is necessary to inactivate them at high temperatures first, or else the detected value of the fiber content of the sample will be lower than the actual value.
- Thank you very much for your suggestion, I have made changes in the article and you can see the revision
- Crystalline methionine was used in this paper. Based on the present results, our team will follow up with further comparisons of the effects of encapsulated methionine, methionine hydroxyl analogs, and methionine metal complexes and their economic benefits.
- Thank you very much for your suggestion, which has been modified to give the recommended dose in the conclusion.
- We carefully reviewed the formatting requirements of the journals and found that the journals did not specify a specific format for the reference format, but only required consistency, so we chose the more commonly used Numbered format and made sure that it was all consistent.
- I have read your note carefully and I apologize that we have discussed and felt that it would not be appropriate to use the Latin name for the keyword, as this is rare and the default is to use the English name.
For the part about nitrogen fixation in faces, see this paper (Alterations in nutrient digestibility and performance of heat-stressed dairy cows by dietary L-theanine supplementation, https://doi.org/10.1016/j.aninu.2022.08.002), who takes the same sulfuric acid fixation as we do. Sulfuric acid and hydrochloric acid serve the same purpose when just measuring conventional nutrients, the purpose being to form ammonium salts and prevent ammonia loss. The difference, however, is that if some heavy metals are to be measured, the results of the two may differ.
In addition to your suggestions, other reviewers' suggestions have also been revised in the paper.
Finally, thanks again for your help!
Best wishes.

Reviewer 3 Report
Comments and Suggestions for Authors
In this paper, the authors evaluated the effects of methionine on the apparent digestibility of nutrients, antler nutrient composition, rumen fluid amino acid composition, fecal volatile fatty acids and intestinal bacteria in antler-growing sika deer.
Based on randomly adding different levels of methionine to the diets of three groups of four deer at 0 g/day (CON), 4 g/day (LMet) and 6 g/day (HMet), the authors demonstrated that Methionine supplementation significantly increased the apparent digestibility of organic matter, neutral detergent fiber (NDF) and acid detergent fiber (ADF) in the LMet group. The crude protein and collagen protein of antlers were significantly higher in the LMet and HMet groups compared to the CON group and also significantly higher in the HMet group compared to the LMet group, while the calcium content of antlers was significantly lower in the HMet group. Ruminal fluid free amino acid composition was altered in the three groups of sika deer, with significant changes in aspartic acid, citrulline, valine, cysteine, methionine, histidine and proline. At the phylum level, Firmicutes and Bacteroidetes were highest in the rectal microflora. Based on the results of principal coordinate analysis and Adonis analysis, there was a significant difference in the composition of the intestinal flora between the CON and HMet groups.
The experiment was balanced with a sufficient number of male sika deer because of the ethics. The experiment and analysis methods are good, but not very detailed. The manuscript was well written.
A few details to add:
Line 88: Was the food distributed completely ingested? if yes, it must be mentioned, if not, the quantities ingested must be added. Because it is not mentioned in reference [15].
Line 91: Put a reference for the partial collection method.
Line 102: Have the antlers been completely cut? if yes, it must be mentioned, if not, the portion deducted must be indicated.

Author Response
Dear professor
Thank you very much for your patient and careful review of my paper and your pertinent suggestions, which have helped me a lot in optimizing my paper.
In response to your comments, the response is as follows
- We determined the feeding level of the sika deer during the pre-test, and therefore arranged a reasonable amount of food during the formal test to ensure that it could be completely consumed by the animal. I've made changes in the paper to indicate that the food was eaten, which you can view in revision mode.
- Thank you very much for your advice, have added some references to the paper to support my collection methodology.
- The antlers were fully captured. I have already stated this in my paper.
In addition to your suggestions, other reviewers' suggestions have also been revised in the paper.
Finally, thanks again for your help!
Best wishes.

Reviewer 4 Report
The reviewers found this paper with interest, showing that adequate methionine supplementation is beneficial in increasing feed efficiency and improving horn quality in Sika deer. Please refer to the comments below.
Major comments
What is the appropriate level of methionine supplementation for sika deer? Is the 4g/day using this time appropriate, and is 6g/day excessive? Also, do you have a rationale for the dosage?
Minor comments
1.P2L84 The "one-week pre-experiment" is for acclimatization to the diet of the main test, and is it the same food composition as the main test?
2.P2L86 Please describe the details of the methionine preparation used this time. Are you using something that isn't lumen bypassed?
3.P3L100 Please indicate where local anesthesia was administered.
ï¼”P3L124, 125, 136 Indicate centrifugation in g(gravity), not r(revolutions per minute).
5.P7L225 Figure 2. The asterisk needs an explanation.
6.P10L275 Please cite the literature.
Author Response
Dear professor
Thank you very much for your patient and careful review of my paper and your pertinent suggestions, which have helped me a lot in optimizing my paper.
In response to your comments, the response is as follows
- I have revised the conclusion to give 4 g/day as the appropriate recommended level for methionine supplementation. The basis for this is the objective antler quality evaluation in our article, as well as the market value assessment by a merchant at the end of the experiment. The market value of both the LMet and HMet groups was higher than that of the CON group, which was about the same price, but this subjective evaluation could not appear in the paper. Also considering the cost of the additives, 4g/day was recommended at the conclusion. The methionine dose determined during the experimental design was based on the results of our team's preliminary methionine requirement during the growing period and the analysis of the amino acid composition of antler, which were calculated and extrapolated to obtain the possible supplemental dose during the antler-growthing period.
- Consistency of food between pre-trial and formal trials. The pre-test is not only to acclimatize the animals to the food, but also by exploring the amount of feed, it makes it possible to make sure that the animals will finish the food every day during the formal experiment. This has been explained in the article.
- This thesis uses crystalline methionine, and based on this study, our team will follow up with further comparisons of the effects of encapsulated methionine, methionine hydroxyl analogs, and methionine metal complexes and their economic benefits. In addition, it has been annotated in the article to clarify the product information of methionine.
- Changes have been made to the article to clarify the site of anesthesia, please view through the revision mode.
- Calculations have been made to convert revolutions to centrifugal force.
- Thank you for reminding me that asterisked notes have been added.
- Reference has been made in the text to figure 4, which corresponds to the results.
In addition to your suggestions, other reviewers' suggestions have also been revised in the paper.
Finally, thanks again for your help!
Best wishes.
